# Fast detection of dam zone boundary based on Otsu thresholding optimized by enhanced harris hawks optimization

**Xiaofeng Qu, Jiajun Wang** [ORCID]**\*, Xiaoling Wang, Yike Hu, Tianwen Tan, Dong Kang**

State Key Laboratory of Hydraulic Engineering Simulation and Safety, Tianjin, 300072, China

\* jiajun_2014_bs@tju.edu.cn

**Data Availability Statement:** The data set and code which constitutes the minimal data set and can be used to reproduce and validate our results is made available in a public repository: https://

## Abstract

Earth-rock dams are among the most important and expensive infrastructure projects. A key safety issue is dam zone boundary detection to prevent the intrusion of materials from different zones. However, existing detection methods strongly highly depend on human judgement, which is time consuming and labor intensive. To solve this problem, this work proposes a fast boundary detection method based on the Otsu algorithm optimized by enhanced Harris hawks optimization (HHO). Compared with the original Otsu algorithm, the proposed method has a higher computation speed to meet the time requirements of engineering projects. Particle swarm optimization is adopted to enhance the exploration stage of HHO. In addition, a tangent function and chaotic sine map are used to improve the convergence speed and robustness. The application of the proposed method to a real-life project shows that the calculation time can be reduced to 20 s, which is approximately 18.8% of the original calculation time.

## 1. Introduction

The construction quality of earth-rock dams is crucial for ensuring the safety of the life and property of downstream personnel. An earth rock dam is divided into a core wall zone, inverted filter zone, transition zone, and rockfill zone. Granular soil materials are used in the core wall zone, and stones of different grades are used in other zones. The construction quality of the core wall zone, with its main anti-seepage effect, is key to quality monitoring. If the materials used in the different zones are not paved in strict accordance with design standards, the intrusion of materials from other zones will lead to poor quality and become a potential safety hazard. Therefore, it is essential to rapidly and accurately detect the boundary of dam zones. However, the rapid detection of material boundaries is difficult in practical engineering applications. Dam zone boundary detection belongs to the category of the edge detection of materials. In general, there are three methods for the edge detection of materials: manual detection, machine learning, and threshold segmentation [1–3]. The conventional paving boundary detection method mainly depends on manual field judgment, which has numerous disadvantages such as human subjectivity and labor intensiveness. Machine learning requires a large training set with abundant samples to train a classification model. If the category

github.com/Feng1673/Plos-one.git. This work uses 3 main data sources: (i) PTSHHO-Otsu, (ii) AHA, (iii) GNDO. The (i) PTSHHO-Otsu case reports data are provided within the repository. The (ii) AHA and the (iii) GNDO are comparative algorithms.

**Funding:** This research was funded by the Yalong River Joint Funds of the National Natural Science Foundation of China (grant number U1965207) and the Young Scientists Fund of China (grant number 52009089).The funders had no role in study design, data collection and analysis, decision to publish, or preparation of the manuscript.

**Competing interests:** The authors have declared that no competing interests exist.

changes, the training set must be reset and relabeled, which is time-consuming and labor intensive. In comparison, threshold segmentation has strong universality, and it has become the preferred method for engineering applications.

Conventional threshold segmentation methods, such as the Otsu algorithm [4], can calculate a paving boundary. However, the calculation time is too long to meet the actual requirements of a project [5,6]. The acceleration of threshold segmentation is an optimization problem [7,8]. Optimization problems have been widely studied in practical engineering applications [9]. The swarm intelligence algorithm has significant computing power and efficiency, and it can overcome the disadvantage of the Otsu algorithm [10]. Swarm intelligence simulates the various group behaviors of social insects or animals and uses the information interaction and cooperation among individuals in a group to achieve optimization. In recent years, numerous swarm optimization algorithms have been proposed and widely applied, such as particle swarm optimization(PSO) [11,12], grey wolf optimization [13], and Harris hawks optimization(HHO) [14]. Heidari et al. [14] compared HHO with other algorithms and demonstrated its advantages. The superiority of HHO has been demonstrated in numerous applications [15–17]. Therefore, this study uses HHO to optimize the Otsu algorithm to improve the speed while ensuring accuracy. However, the accuracy of HHO is significantly affected by population initialization, and the convergence speed and accuracy cannot meet engineering requirements. In addition, HHO has the disadvantage that it falls into local optimization. Therefore, to solve the problem of population initialization, this study uses the chaotic sine map to optimize the initial population. Owing to the low convergence speed and insufficient convergence accuracy, the tanh activation function is used to optimize the escape energy of prey (threshold). The PSO is used to optimize the exploration stage to address the problem of local optimization.

## 2. Engineering optimization problems

This study focuses on the detection of paving boundaries in the core wall area of earth-rock dams. Images are mainly obtained using the monitoring cameras on the dam abutments on both banks. The conventional Otsu calculation of paving boundaries is time consuming and cannot meet the requirements of field engineering. There is an urgent requirement for an optimization algorithm to meet the recognition accuracy of paving boundaries in complex environments and significantly improve the calculation time.

## 3. Methodology

Otsu is one of the most advanced methods in the field of threshold segmentation. Otsu assumes that there is a threshold that divides all the pixels of an image into two categories. Each calculation traverses all 256 gray values to determine the optimal threshold, leading to a large number of calculations. Therefore, this study uses HHO to simplify the calculation of Otsu and rapidly determine the optimal threshold. Then, a chaotic sine map is used for optimization to eliminate the dependence of the HHO algorithm on the initial population. Finally, the speed update logic of PSO is used to resolve the problem of HHO falling into local optimization. Additionally, the escape energy is optimized using the tanh function, which improves the accuracy and convergence speed of the algorithm.

### 3.1 Brief introduction of Otsu

The Otsu algorithm assumes a threshold (k) that divides all pixels into two categories: C1 (less than k) and C2 (greater than k). The average values of C1 and C2 are $m_1$ and $m_2$, respectively, and the global average value of an image is mG. The probabilities of pixels being divided into

C1 and C2 are $P_1$ and $P_2$, respectively. Therefore, there are:

$$P_1 * m_1 + P_2 * m_2 = mG \tag{1}$$

$$P_1 + P_2 = 1 \tag{2}$$

According to the concept of variance, the expression of the variance between classes is as follows:

$$\sigma^2 = P_1(m_1 - mG)^2 + P_2(m_2 - mG)^2 \tag{3}$$

Where $\sigma^2$ is the variance.

On the basis of traversing 0–255 gray levels, k is the best threshold that makes Eq (3) reach a maximum. This process requires a significant amount of computation.

Otsu has also been optimized using various targeted methods [18–21]. After the two-dimensional (2D) Otsu algorithm was proposed, it rapidly replaced the original Otsu algorithm and was widely used because of its superior antinoise performance. 2D Otsu adds a one-dimensional threshold parameter (average gray level of the neighborhood) to the original Otsu algorithm [Eq (4)]. This study uses 2D Otsu

$$\sigma_{k,t}{}^2 = P_1(k,t)(m_1(k,t) - mG)^2 + P_2(k,t)(m_2(k,t) - mG)^2 \tag{4}$$

The ergodicity of Otsu ensures its accuracy but also considerably increases the calculation time, which cannot meet project time requirements. Therefore, this study proposes the optimization of the Otsu algorithm with enhanced HHO to ensure the accuracy of detection while reducing the calculation time.

## 3.2 Proposed HHO enhanced by tanh and PSO

HHO is a population optimization algorithm proposed by Heidari in 2019 [14]. The algorithm simulates the predatory behavior of Harris hawks (raptors in southern Arizona, USA), and it is divided into an exploration stage, exploration-development transformation stage, and development stage. Table 1 shows that HHO can find the optimal solution relatively better than AHA [22] and GNDO [23]. However, it falls into the local optimal solution in a few cases. This study optimizes the exploration and exploration-development transformation stages and the population initialization.

1) Exploration stage optimized by PSO

In HHO, each Harris hawk is a candidate, and the best candidate in each process is evaluated to locate the expected prey or be close to the expected prey. Harris hawks randomly perch at certain positions and wait to detect prey according to two strategies. We consider that the opportunity q of each strategy is equal. When q < 0.5, a Harris hawk perches based on the position of other members and prey. When q ≥ 0.5, the Harris hawk randomly perches on the large tree within the range of the hawk group, and the specific model is as follows:

$$X(t+1) = \begin{cases} X_{\text{rand}}(t) - r_1|X_{\text{rand}}(t) - 2r_2X(t)|, q \geq 0.5 \\ (X_{\text{rabbit}}(t) - X_m(t)) - r_3(LB + r_4(UB - LB)), q < 0.5 \end{cases} \tag{5}$$

Where X (t + 1) is the position vector of eagle in the next iteration, X $_{\text{rabbit}}$ (t) is the position of the prey (i.e., the position of the individual with the best fitness), X (t) is the position vector of the current eagle, $r_1$, $r_2$, $r_3$, $r_4$ and q are random numbers in (0,1), LB and UB are lower and upper bounds of variables, X $_{\text{rand}}$ is the randomly selected position of the hawk in the current

**Table 1. Comparison of HHO with AHA and GNDO.**

| Function | Name | d | $f_{min}$ | HHO | AHA | GNDO |
|---|---|---|---|---|---|---|
| $f_1(x)$ | Sphere Function | 30 | 0 | 8.67E-84 | 1.54E-146 | **8.32E+03** |
| $f_2(x)$ | Schwefel's Prolem2.22 | 30 | 0 | 2.07E-43 | 5.79E-66 | **3.04E+01** |
| $f_3(x)$ | Schwefel's Prolem1.2 | 30 | 0 | 1.27E-51 | 7.59E-112 | **1.19E+04** |
| $f_4(x)$ | Schwefel's Prolem2.21 | 30 | 0 | 3.45E-40 | 1.43E-56 | **3.83E+01** |
| $f_5(x)$ | Generalized Rosenbrock's Function | 30 | 0 | 8.66E-02 | **2.79E+01** | **3.75E+06** |
| $f_6(x)$ | Step Function | 30 | 0 | 5.63E-03 | **1.15E+00** | **8.68E+03** |
| $f_7(x)$ | Quartic Function i.e.niose | 30 | 0 | 3.89E-04 | 5.92E-04 | **3.17E+00** |
| $f_8(x)$ | Generalized Schwefel's Prolem2.26 | 30 | −418.9829×n | -1.26E+04 | **-1.01E+04** | **-5.67E+03** |
| $f_9(x)$ | Generalized Rastrigin's Function | 30 | 0 | 0.00E+00 | 0.00E+00 | **1.25E+02** |
| $f_{10}(x)$ | Ackley Function | 30 | 0 | 8.88E-16 | 8.88E-16 | **1.51E+01** |
| $f_{11}(x)$ | Generalized Griewank's Function | 30 | 0 | 0.00E+00 | 0.00E+00 | **6.56E+01** |
| $f_{12}(x)$ | Generalized Penalized Function(1) | 30 | 0 | 6.64E-05 | 2.05E-02 | **1.81E+06** |
| $f_{13}(x)$ | Generalized Penalized Function(2) | 30 | 0 | 3.93E-04 | 2.29E+00 | **1.16E+07** |
| $f_{14}(x)$ | Shekel's Foxholes Function | 2 | 1 | **1.64E+00** | 1.10E+00 | **2.38E+00** |
| $f_{15}(x)$ | Kowalik's Function | 4 | 0.00030 | **4.99E-04** | **4.15E-04** | **5.79E-03** |
| $f_{16}(x)$ | Six-Hump Camel-Back Function | 2 | -1.0316 | -1.03E+00 | -1.03E+00 | -1.03E+00 |
| $f_{17}(x)$ | Branin Function | 2 | 0.398 | 3.98E-01 | 3.98E-01 | 3.98E-01 |
| $f_{18}(x)$ | Goldstein-Price Function | 2 | 3 | 3.00E+00 | 3.00E+00 | 3.00E+00 |
| $f_{19}(x)$ | Hartman's Family(d = 3) | 3 | -3.86 | -3.85E+00 | -3.86E+00 | -3.86E+00 |
| $f_{20}(x)$ | Hartman's Family(d = 6) | 6 | -3.32 | -2.94E+00 | -3.30E+00 | -3.16E+00 |
| $f_{21}(x)$ | Shekel's Family(m = 5) | 4 | -10.1532 | **-5.50E+00** | **-7.45E+00** | **-6.12E+00** |
| $f_{22}(x)$ | Shekel's Family(m = 7) | 4 | -10.4028 | **-5.29E+00** | **-8.14E+00** | **-6.95E+00** |
| $f_{23}(x)$ | Shekel's Family(m = 10) | 4 | -10.5363 | **-5.52E+00** | **-7.74E+00** | **-6.95E+00** |
| Error | | | | 5 | 7 | 18 |

Note: Bold does not indicate convergence, that is, the solution that does not meet the standard. The judgment basis of the standard solution in this work is that the error must be within 20% of $f_{min}$; when $f_{min}$ is 0, the error is 0.2.

population.

$$X_{m}(t) = \frac{1}{N}\sum_{i=1}^{N} X_i(t) \tag{6}$$

Where $X_m(t)$ is the average position of individuals.

The results of the test function show that HHO falls into local optimization in individual cases [24,25]. The speed update logic of PSO is used solve this problem so that HHO can explore more comprehensively in the exploration stage.

The speed update logic of PSO [Eq (7)] ensures the global nature of the population search and significantly reduces the probability of falling into local optimization.

$$v_i^{k+1} = \omega * v_i^k + c_1 r_1 (xbest^k - x_i^k) + c_2 r_2 (xgbest^k - x_i^k) \tag{7}$$

Where: $\omega$ is the inertia factor, whose value is non-negative; $c_1$ and $c_2$ are learning factors in the range [0,4]; $v_i^{k+1}$ and $v_i^k$ are the velocity of the particles; $x_i^k$ is the current position of the particle; $xbest^k$ and $xgbest^k$ indicate the best positions experienced thus far by the ith particle and the whole swarm, respectively; and are two random numbers that are uniformly distributed in the range (0,1).

Based on Eq (7), we improve the part for $q \geq 0.5$, as shown in Eq (5).

$$X(t+1) = \omega * X(t) + c_1 r_1 (X_{\text{rabbit}}(t) - X(t)) + c_2 r_2 (X_{\text{rabbit}}(t) - X_{\text{rand}}(t)) \tag{8}$$

$\omega$ is set as 0.5, $c_1$ and $c_2$ are set as 1, and Eq (9) is obtained.

$$X(t+1) = 0.5 * X(t) + r_1 (X_{\text{rabbit}}(t) - X(t)) + r_2 (X_{\text{rabbit}}(t) - X_{\text{rand}}(t)) \tag{9}$$

2) Exploration development transformation stage optimized by tanh

The HHO algorithm can be transferred from exploration to development, and then, according to the escape energy of the prey, it can be converted to different development behaviors. The energy of the prey is considerably reduced during the escape process. The escape energy of the prey is as follows:

$$E = 2E_0(1 - \frac{t}{T}) \tag{10}$$

Where E is the escape energy of prey, T is the maximum number of iterations, and $E_0$ is the initial state of its energy.

In the iterative process of HHO, $E_0$ randomly varies within (- 1, 1). When $E_0$ decreases from 0 to -1, the prey physically flags, whereas when $E_0$ increases from 0 to 1, the prey activity increases. The dynamic escape energy E decreases during the iteration. When the escape energy $|E|$ is greater than or equal to 1, the hawks search different zones to explore the location of their prey. When $|E|$ is less than 1, the neighborhood of the solution is searched as the algorithm attempts to develop the exploration phase.

In this study, the tanh activation function used in deep learning is introduced to optimize the relationship between E and the number of iterations to improve the overall accuracy.

$$E = 2E_0(1 - \frac{4 * \tanh(\frac{t}{T})}{\pi}) \tag{11}$$

3) Population initialization optimized by chaotic mapping

In 1976, biologist Robert May applied chaotic mapping [26]. Since then, chaotic mapping has been widely used [27–31]. The design of a chaotic stream cipher system mainly uses the following chaotic maps: a one-dimensional logistic map, 2D Henon map, three-dimensional Lorenz map, piecewise linear chaotic map, and piecewise nonlinear chaotic map. Dehkordi et al. proposed that a chaotic sine map improved the results of HHO more significantly than its counterparts [32]. In this study, we use certain characteristics of sine maps.

**Table 2. Pseudocode.**

| Pseudo code of Otsu Thresholding Optimized by Enhanced HHO |
| --- |
| Inputs: **The population size N, maximum number of iterations T and image** |
| Outputs: **The location of rabbit and its fitness value(variance)** |
| **Use sine mapping to initialize the Population (** Eq (12)) |
| while **(stopping condition is not met) do** |
| **Calculate the fitness values of hawks** |
| Set Xrabbit as the location of rabbit (best location) |
| Calculate the probability of gray value |
| for **(each hawk (Xi)) do** |
| **Update the initial energy E0 and jump strength J** |
| E0 = 2rand()-1, |
| J = 2(1-rand()) |
| **Update the E using Eq (11)** |
| **E = 2*(1–4*tanh(t/T)/pi)*E0** |
| if ($|E| \geq 1$) |
| **Update the location vector using Eq (5) And Eq (9)** |
| if ($|E| < 1$) |
| if (r $\geq$ 0.5 and $|E| \geq$ 0.5) |
| **Update the location vector** |
| else if (r $\geq$ 0.5 and $|E| <$ 0.5) |
| **Update the location vector** |
| else if (r $<$ 0.5 and $|E| \geq$ 0.5) |
| **Update the location vector** |
| else if (r $<$ 0.5 and $|E| <$ 0.5) |
| **Update the location vector** |
| Return Xrabbit |
| Return the location of rabbit and its fitness value |

The mathematical expression of sine mapping is as follows:

$$X_{n+1} = \frac{a}{4}\sin(\pi * X_n) \tag{12}$$

where a$\in$ [0,4] is called logistic parameter. It is shown that when $X_n \in$ [0,1], the sine mapping is in chaos. $X_n$ can achieve ergodicity when a = 4. In this study, a is set to 4.

4) Pseudocode of Otsu thresholding optimized by enhanced HHO

The pseudo code of Otsu Thresholding Optimized by Enhanced HHO is shown in Table 2. The specific code is in data availability.

5) Flowchart of Otsu thresholding optimized by enhanced HHO

Fig 1 is the flowchart of Otsu thresholding optimized by enhanced HHO. Through the flowchart, we can quickly understand the calculation process and the specific location of optimization.

## 3.3 Performance testing of enhanced HHO

The effectiveness of the proposed enhanced HHO, is tested using a set of well-studied benchmark functions from existing literature [33–35] similar to the original HHO [14]. This set consists of two main groups: unimodal (UM) and multimodal (MM). The UM functions (F1 –F7) with unique global optimization can determine the development (intensification) capabilities of different optimizers, and the MM functions (F8 –F23) can determine the exploration (diversification) of algorithms and the potential of LO avoidance. Composition is not selected becausethe problems considered in this study are limited to UM and MM cases.

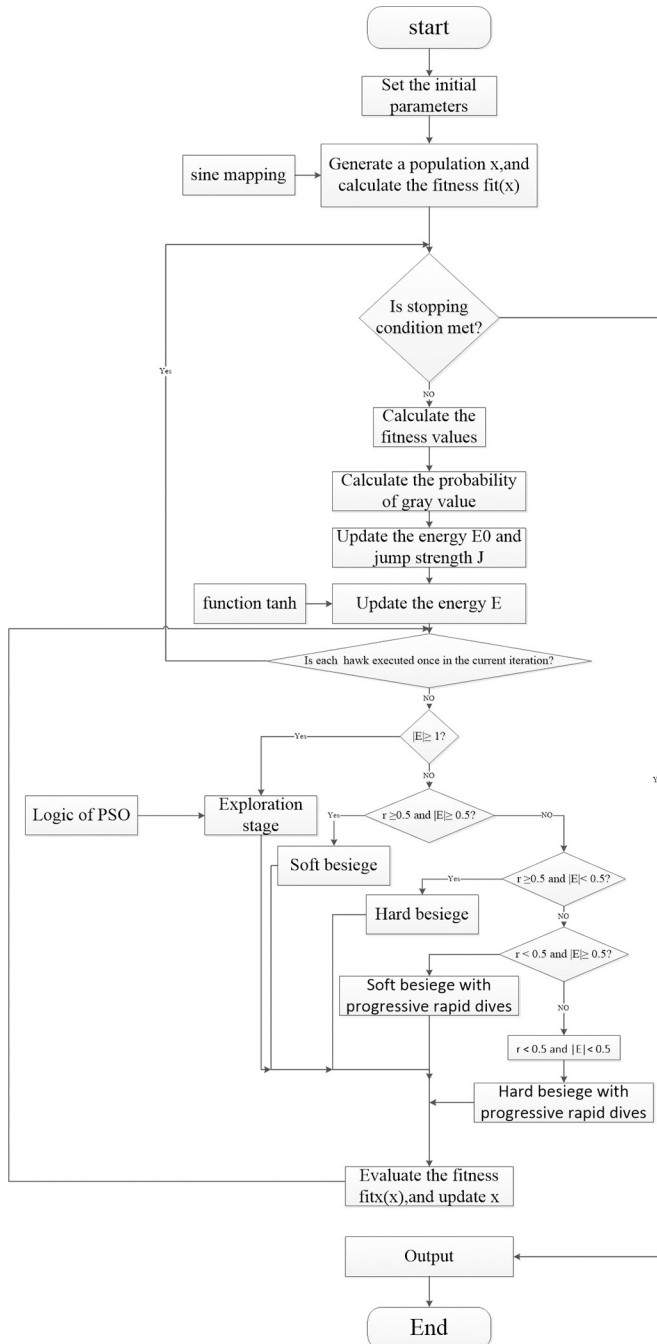

**Fig 1. Flowchart of Otsu thresholding optimized by enhanced HHO.**

The average of results (AVG) and standard deviation (STD) indices, which are commonly used in multiple HHO studies, are used for judgment (Dehkordi, et al., 2021; Krishna, Saxena, & Kamboj, 2021; Qu, et al., 2020). All the models adopt the same conditions with a population of 10 and 500 iterations, considering the corresponding engineering application requirements.

The algorithm whose original population is optimized using the sine map is referred to as SHHO. The algorithm that optimizes the escape energy of SHHO using the tanh activation

**Table 3. Unimodal benchmark functions.**

| Function | Name | d | Range | $f_{min}$ | AVG | STD |
|---|---|---|---|---|---|---|
| $f_1(x)$ | Sphere Function | 30 | [−100,100] | 0 | $3.10 \times 10^{-89}$ | $1.12 \times 10^{-88}$ |
| $f_2(x)$ | Schwefel's Prolem2.22 | 30 | [−10,10] | 0 | $2.92 \times 10^{-46}$ | $1.25 \times 10^{-45}$ |
| $f_3(x)$ | Schwefel's Prolem1.2 | 30 | [−100,100] | 0 | $9.09 \times 10^{-59}$ | $2.98 \times 10^{-58}$ |
| $f_4(x)$ | Schwefel's Prolem2.21 | 30 | [−100,100] | 0 | $3.37 \times 10^{-46}$ | $1.05 \times 10^{-45}$ |
| $f_5(x)$ | Generalized Rosenbrock's Function | 30 | [−30,30] | 0 | $5.80 \times 10^{-02}$ | $8.89 \times 10^{-02}$ |
| $f_6(x)$ | Step Function | 30 | [−100,100] | 0 | $9.41 \times 10^{-04}$ | $1.20 \times 10^{-03}$ |
| $f_7(x)$ | Quartic Function i.e.niose | 30 | [−1.28,1.28] | 0 | $3.36 \times 10^{-04}$ | $2.00 \times 10^{-04}$ |

function is referred to as TSHHO. The algorithm that optimizes the location update strategy of the exploration stage of TSHHO using the speed update of PSO is referred to as PTSHHO. Tables 3 and 4 present the test results for the UM and MM functions, respectively (test results of PTSHHO).

**3.3.1 AVG and STD for PTSHHO.** This section explains the advantages of PTSHHO using two indicators: AVG and STD. Fig 2 shows the results of 20 tests for the UM problems (F1–F7) obtained using different optimized HHO algorithms. The optimal solutions of F1–F7 are all 0, so the smaller the value, the better. The red line is the standard deviation and the

**Table 4. Multimodal benchmark functions.**

| Function | Name | d | Range | $f_{min}$ | AVG | STD |
|---|---|---|---|---|---|---|
| $f_8(x)$ | Generalized Schwefel's Prolem2.26 | 30 | [−500,500] | −418.9829×n | $-1.26 \times 10^4$ | 4.77 |
| $f_9(x)$ | Generalized Rastrigin's Function | 30 | [−5.12,5.12] | 0 | 0.00 | 0.00 |
| $f_{10}(x)$ | Ackley Function | 30 | [−32,32] | 0 | $8.88 \times 10^{-16}$ | 0.00 |
| $f_{11}(x)$ | Generalized Griewank's Function | 30 | [−600,600] | 0 | 0.00 | 0.00 |
| $f_{12}(x)$ | Generalized Penalized Function(1) | 30 | [−50,50] | 0 | $6.03 \times 10^{-05}$ | $9.03 \times 10^{-05}$ |
| $f_{13}(x)$ | Generalized Penalized Function(2) | 30 | [−50,50] | 0 | $4.28 \times 10^{-04}$ | $6.30 \times 10^{-04}$ |
| $f_{14}(x)$ | Shekel's Foxholes Function | 2 | [−65,65] | 1 | 1.15 | 0.364 |
| $f_{15}(x)$ | Kowalik's Function | 4 | [−5,5] | 0.00030 | $3.73 \times .7^{-04}$ | $5.35 \times .3^{-05}$ |
| $f_{16}(x)$ | Six-Hump Camel-Back Function | 2 | [−5,5] | -1.0316 | -1.03 | 0.00 |
| $f_{17}(x)$ | Branin Function | 2 | [−5,5] | 0.398 | 0.398 | $1.58 \times 10^{-04}$ |
| $f_{18}(x)$ | Goldstein-Price Function | 2 | [−2,2] | 3 | 3.00 | $5.23 \times 10^{-05}$ |
| $f_{19}(x)$ | Hartman's Family(d = 3) | 3 | [0,1] | -3.86 | -3.86 | $3.64 \times .6^{-03}$ |
| $f_{20}(x)$ | Hartman's Family(d = 6) | 6 | [0,1] | -3.32 | -3.01 | -3.01 |
| $f_{21}(x)$ | Shekel's Family(m = 5) | 4 | [0,10] | -10.1532 | -8.70 | 2.19 |
| $f_{22}(x)$ | Shekel's Family(m = 7) | 4 | [0,10] | -10.4028 | -8.37 | 2.51 |
| $f_{23}(x)$ | Shekel's Family(m = 10) | 4 | [0,10] | -10.5363 | -9.14 | 2.10 |

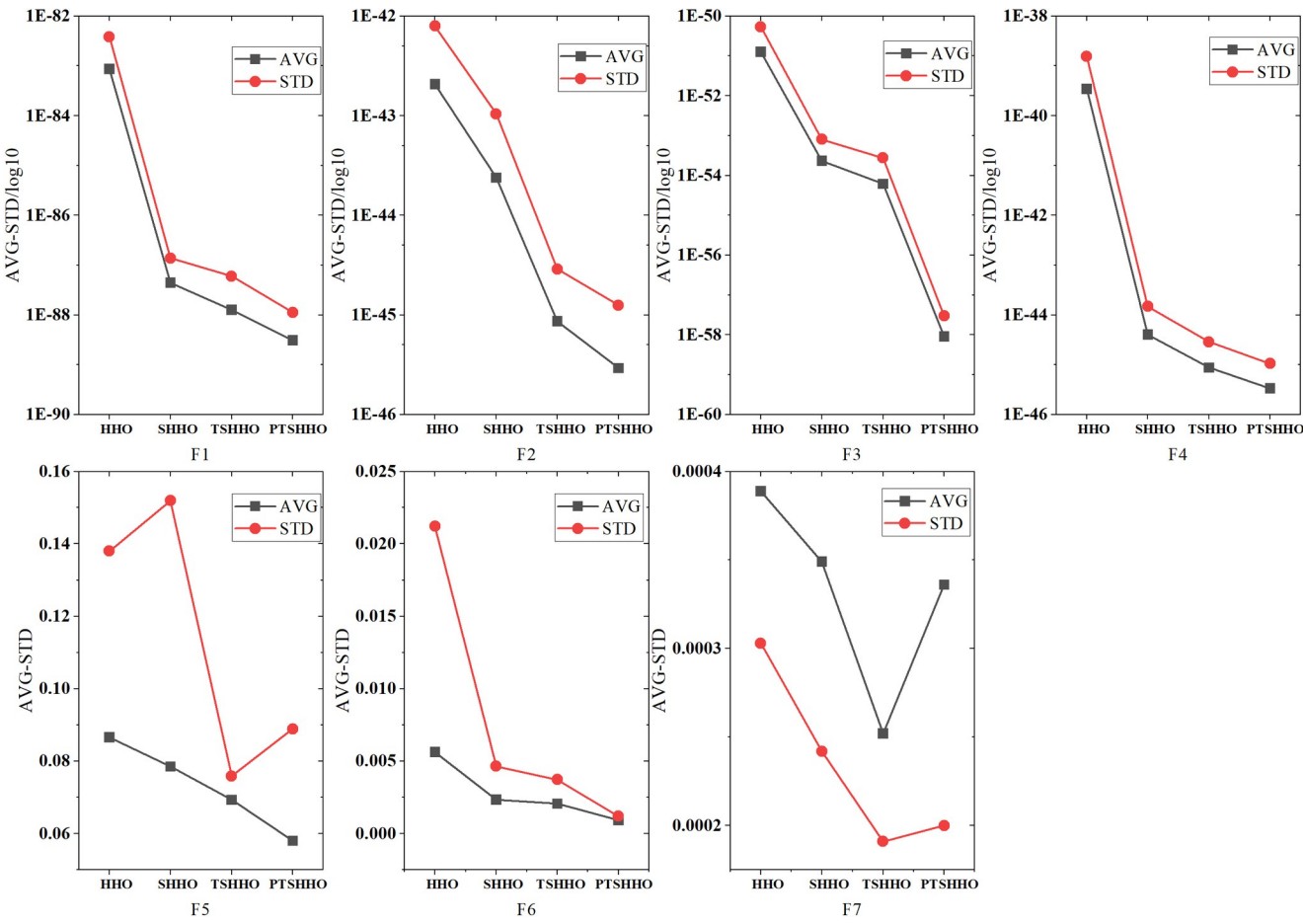

**Fig 2. Results of unimodal benchmark functions (F1–F7).**

black line is the average. In most cases, PTSHHO is optimal for UM problems (F1-F6) and TSHHO is optimal for a few subproblems (F7). PTSHHO and TSHHO are better than SHHO and the original HHO.

Fig 3 shows the results of 20 tests for the MM problems (F8–F23) obtained using different optimized HHO algorithms. The calculation results of F9–F11 and F16 for different HHO algorithms obtain the optimal solution (thus, there is no chart description in this study), indicating the consistency of the results of the different methods. Combined with $f_{min}$ in Table 4, PTSHHO can achieve the best results except F8, F12 and F13. Except PTSHHO, the other three HHO algorithms typically fall into local optimization, which only reaches half of the extreme value. PTSHHO does not fall into local optimization. However, among the 20 tests, the solution still falls into local optimization in a few cases (which may be limited by the population and number of iterations).

In summary, PTSHHO provides the advantage of consistent accuracy in the UM and MM problems. In addition, PTSHHO has a stronger ability to move out of the local optimal solution to obtain the global optimum. This provides a fast and efficient method of meeting the time requirements of engineering projects.

**3.3.2 Convergence curve of PTSHHO.** In this section, we discuss certain phenomena observed during the convergence behavior of PTSHHO for the UM and MM problems. The

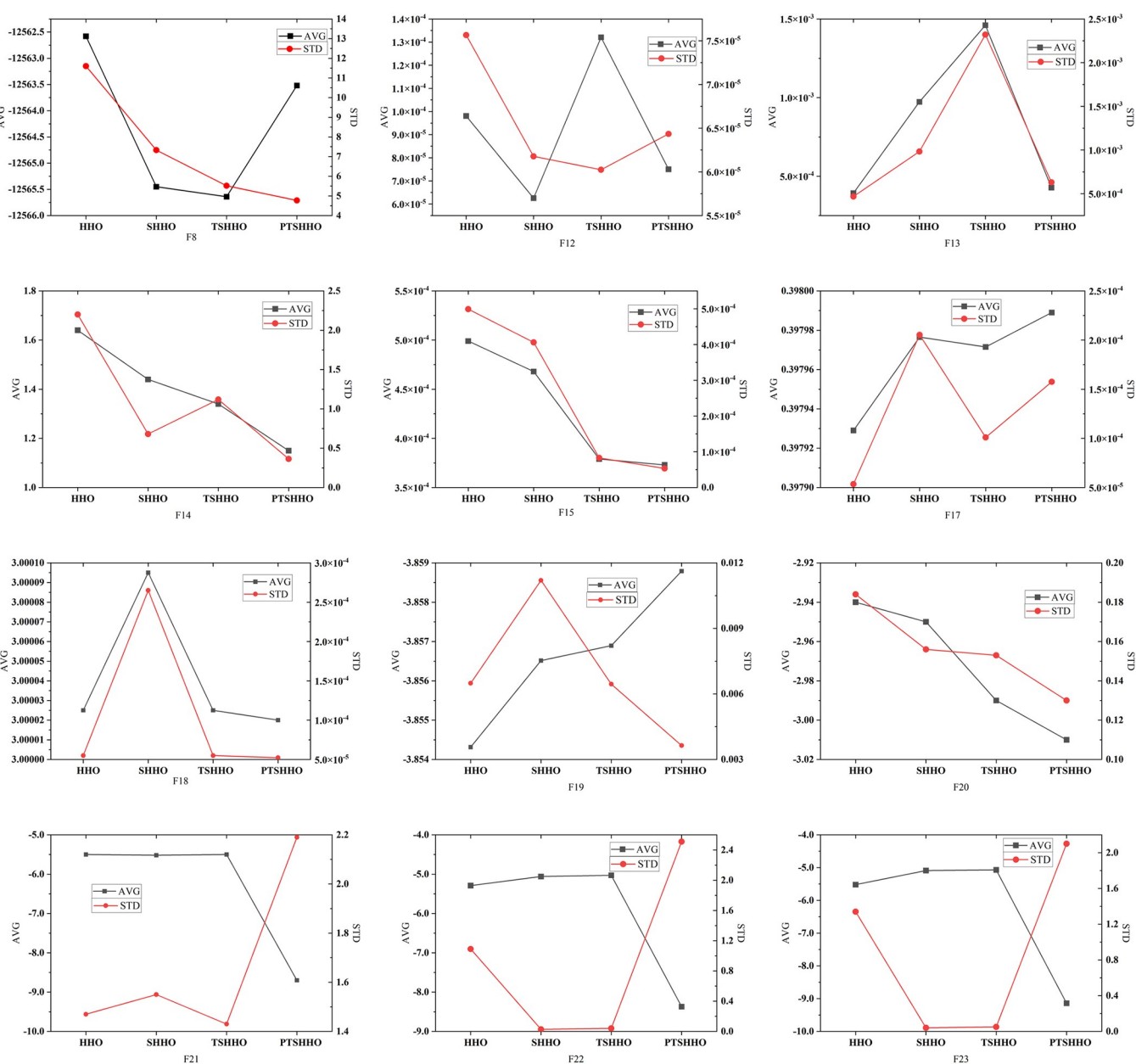

**Fig 3. Results of multimodal benchmark functions.**

convergence behavior is in the form of a curve, and it directly reflects the convergence speed and accuracy.

Fig 4 shows the convergence behavior for the UM problems and the average results of the 20 tests. It is worth noting that in order to make GNDO converge more easily, F1-F7 corresponding to GNDO in Fig 4 are the test data when dim = 2, and the convergence curves of other algorithms are tested when dim = 30. Fig 4 shows that PTSHHO is more stable than AHA and GNDO. There will be no problem of falling into local optimization in unimodal test function, which is more in line with the application of on-site complex environment. PTSHHO and TSHHO converge considerably faster for F1–F4. In addition, PTSHHO has

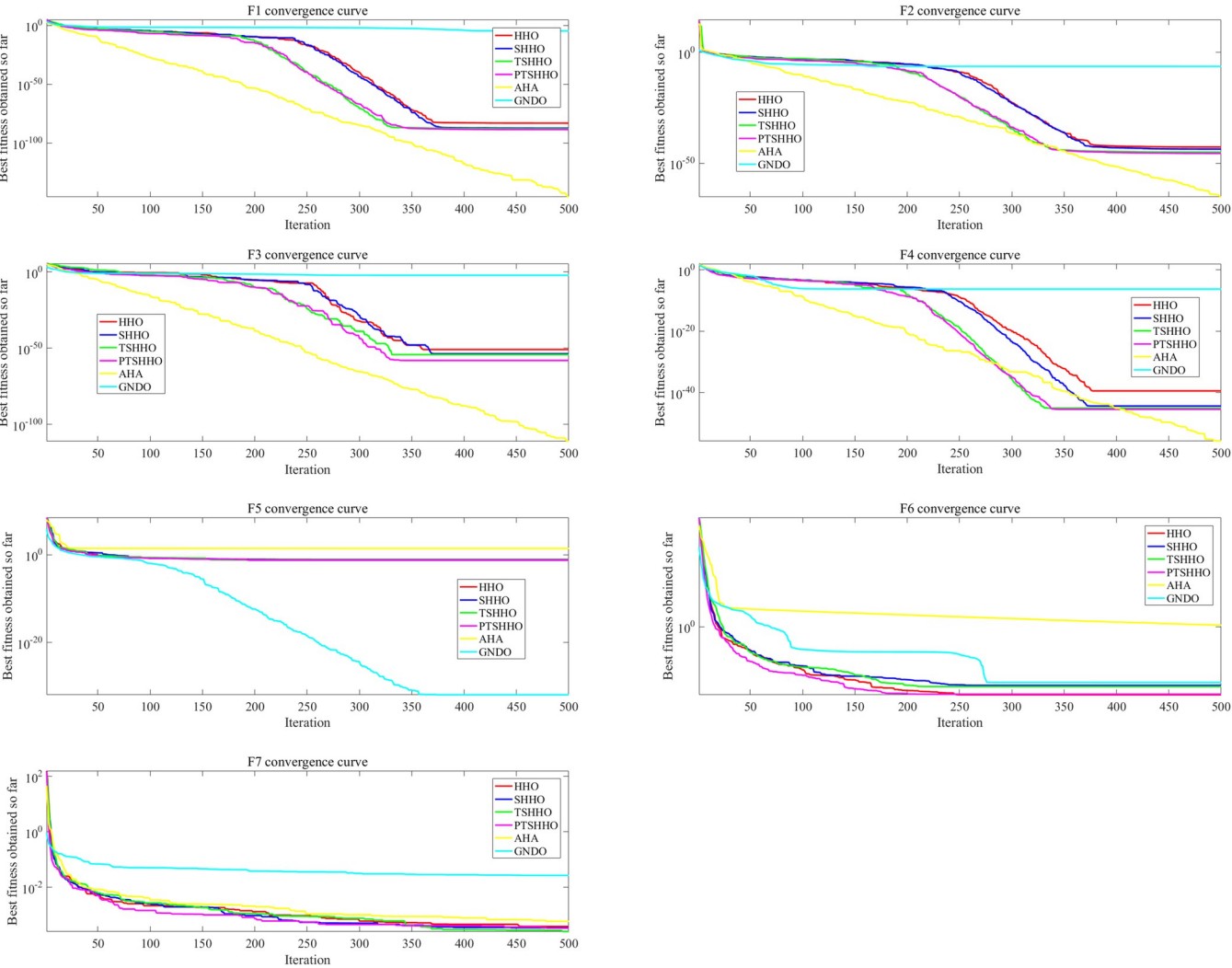

**Fig 4. Convergence behavior for unimodal problems.**

higher accuracy, particularly for F3. The convergence speed of PTSHHO is relatively fast but not evident for F5–F7. The convergence accuracy of PTSHHO is slightly better for F5–F6 but not for F7. In summary, PTSHHO shows a higher convergence speed for the UM problems and better convergence accuracy in most cases.

In the case of the MM problems, there is a gratifying phenomenon in the convergence behavior of PTSHHO. The results for the three MM problems shown in Fig 5 are unique. All algorithms except PTSHHO fall into local optimal solutions. As the convergence accuracy varies significantly compared to the previous part of the convergence curve, PTSHHO has a higher convergence speed.

The following conclusions can be drawn from the convergence curves for the UM and MM problems. PTSHHO provides a higher convergence speed. The number of iterations can be reduced for different problems. This can effectively reduce the calculation time to meet the time requirements of engineering projects.

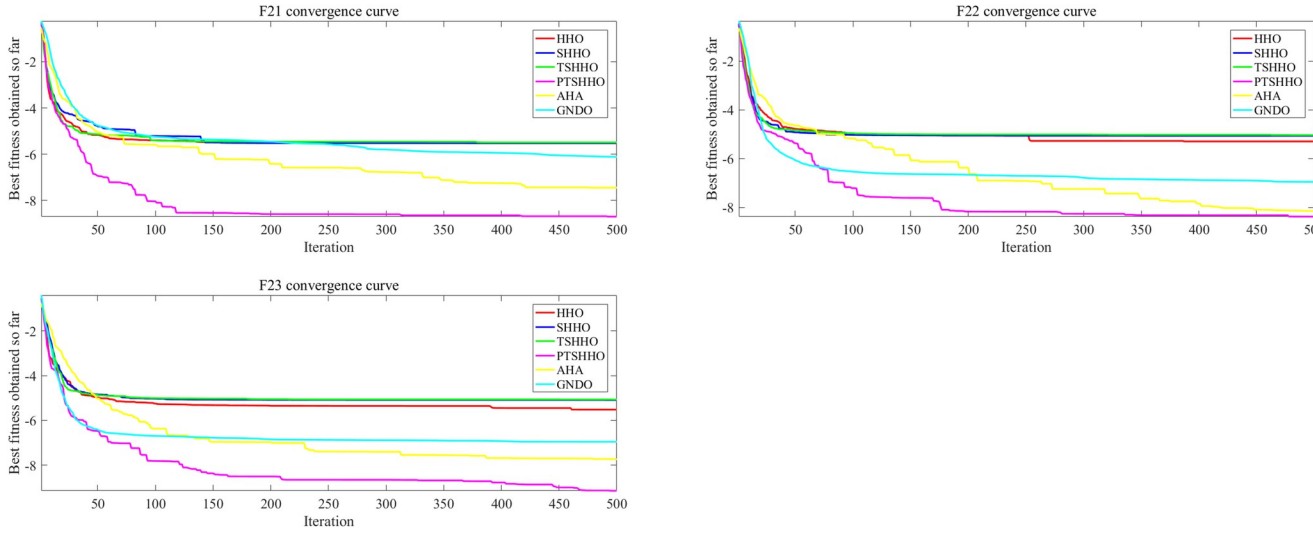

**Fig 5. Convergence behavior for multimodal problems (F21-F23).**

## 4. Case study

A practical construction project is used to evaluate the performance of the rapid dam zone boundary detection method based on the Otsu threshold optimized using the enhanced HHO. The data are mainly obtained from the large-scale high earth rock dam project (LHK), which is under construction in Southwest China. Section 4.1 briefly describes the data acquisition and preprocessing. Section 4.2 describes the timeliness and accuracy of the proposed method. Section 4.3 presents the actual application of the method in the field.

### 4.1 Data acquisition and preprocessing

The engineering data are collected from LHK. Its maximum dam height elevation is 295 m, ranking first among dams of similar types worldwide. Images are obtained from ivms-8700 installed on the dam abutments. In addition, the image data are logically mapped with the actual coordinates to ensure that the image can be fed back to the scene on time after rapid detection.

Fig 6 shows the images that are simultaneously collected at the dam abutments on the left and right banks to ensure that the entire zone surface is clearly recorded. The cameras on these dam abutments collect one image every minute and save it to check and accept the paving quality using the proposed method after paving on the zone surface.

Fig 7 shows the entire process of the basic method, from the paving process to quality acceptance. First, the zone surface information is extracted from the earth rock dam rolling real-time monitoring system developed by our research group [36]. Second, the corresponding image is obtained from the dataset collected by the camera, and the zone surface to be constructed is extracted. Finally, the paving boundary in the target area is identified and compared with the design paving boundary read by the paving system. If paving encroachment occurs, it is fed back to the corresponding onsite personnel according to different alarm levels, and they provide feedback.

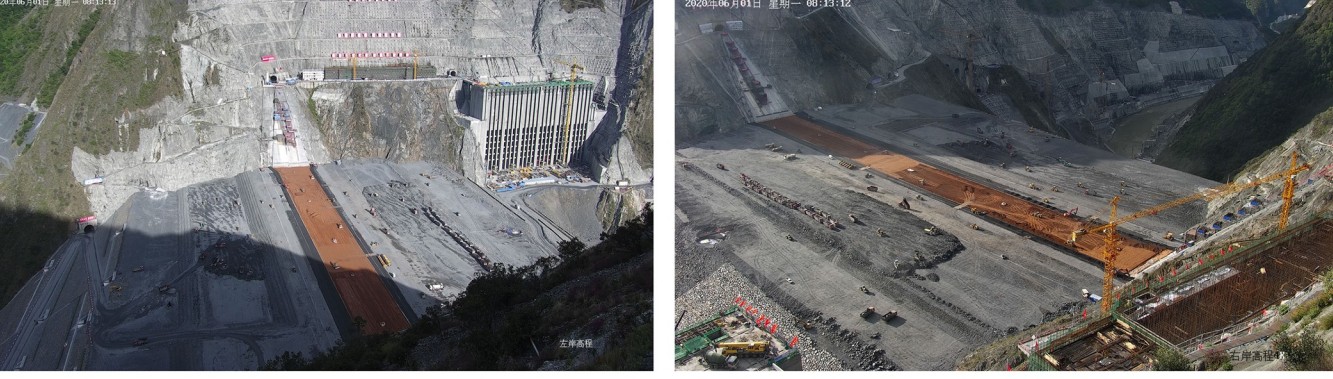

**a:Left bank abutment**          **b:Right bank abutment**

**Fig 6. Image collected from bank abutment.**

## 4.2 Timeliness and accuracy of the method

The improved HHO is used to optimize the calculation speed of 2D-Otsu, and $\sigma^2$ in Eq (8) is rapidly obtained. In the test problems the optimal solution is the minimum value, whereas the

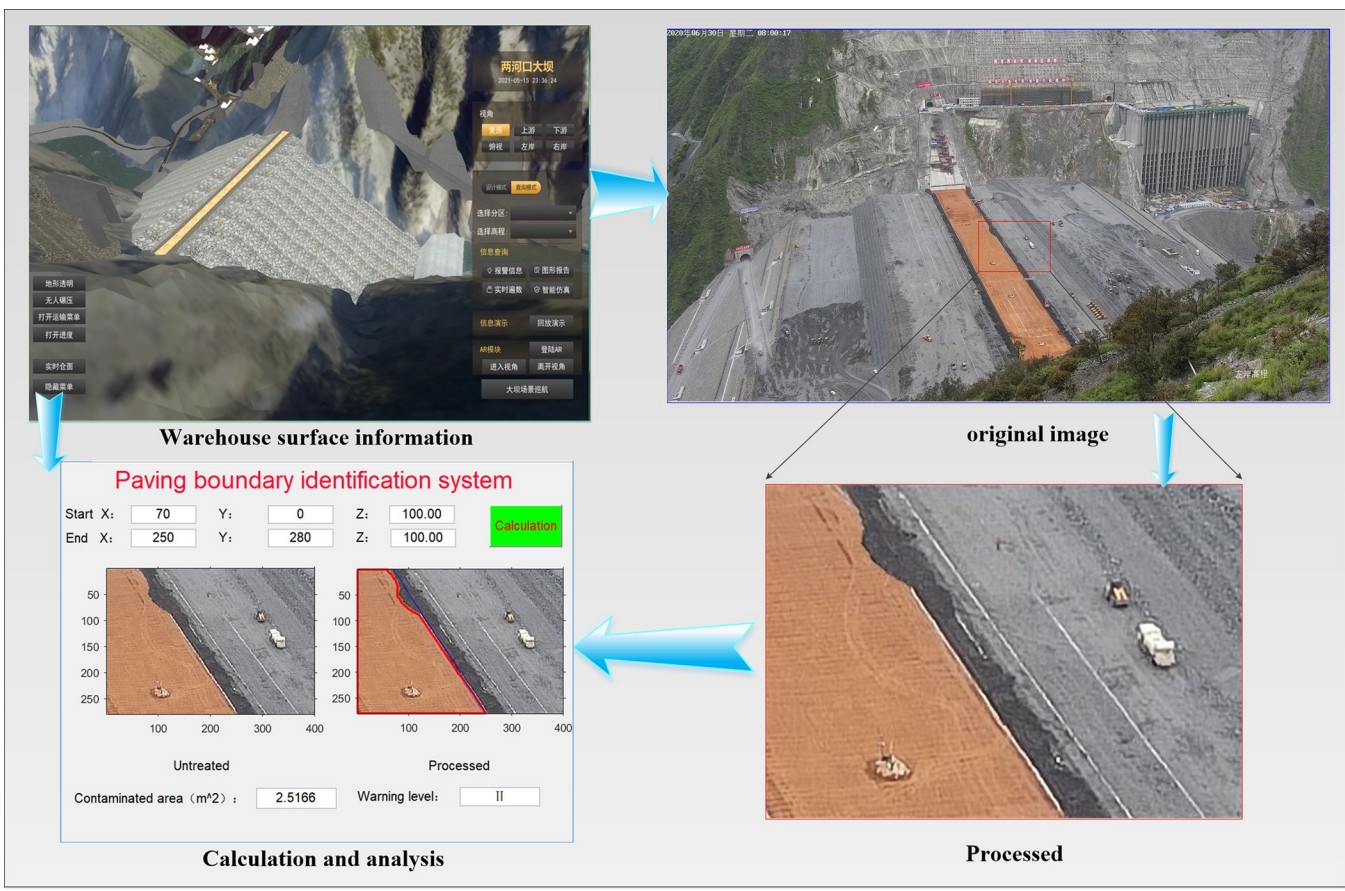

**Fig 7. Process of the application system.**

**Table 5. Practical engineering case results.**

|  | Metric | 2D original Otsu | HHO -Otsu | SHHO -Otsu | TSHHO -Otsu | PTSHHO -Otsu |
|---|---|---|---|---|---|---|
| $\sigma^2$ | AVG | 716.5535 | 716.4502 | 716.4502 | 716.4502 | 716.5535 |
|  | STD | 0.0000 | 0.4621 | 0.4621 | 0.4621 | 0.0000 |
| Time | AVG | 104.0427 | 20.89542 | 20.2016 | 21.9968 | 19.5909 |
|  | STD | 11.1917 | 1.7707 | 2.7606 | 3.0011 | 0.6712 |

target for 2D-Otsu is the maximum value. The zone surface selected in Fig 6 is considered as a case study.

Table 5 shows the results of the actual engineering cases. The data in Table 5 are the average of 20 tests. The calculation time of the Otsu algorithm improved by different HHO algorithms is significantly lower than that of the original 2D Otsu algorithm. The calculation time of the original Otsu algorithm is approximately 5 times that of the improved Otsu algorithm. The algorithms other than PTSHHO have the disadvantage of falling into local optimization. In general, PTSHHO can obtain the optimal solution with the shortest calculation time.

Fig 8 shows the convergence curve of a case for various optimization algorithms. The convergence speed of the four optimized algorithms is quite high, among which PTSHHO-Otsu converges the fastest in the early stage and can obtains the extreme value.

## 4.3 Field application

Engineering data are collected from LHK, and the system based on PTSHHO-Otsu is applied to the project. This effectively reduces the labor cost and time required for the corresponding processes.

Fig 9 shows the system applied at the project site and the paving results of the zone surface. The coarse aggregate occupies the core wall zone, and a first-class early warning is generated when the zone area is more than 4 m$^2$. A level II early warning is generated when the zone area

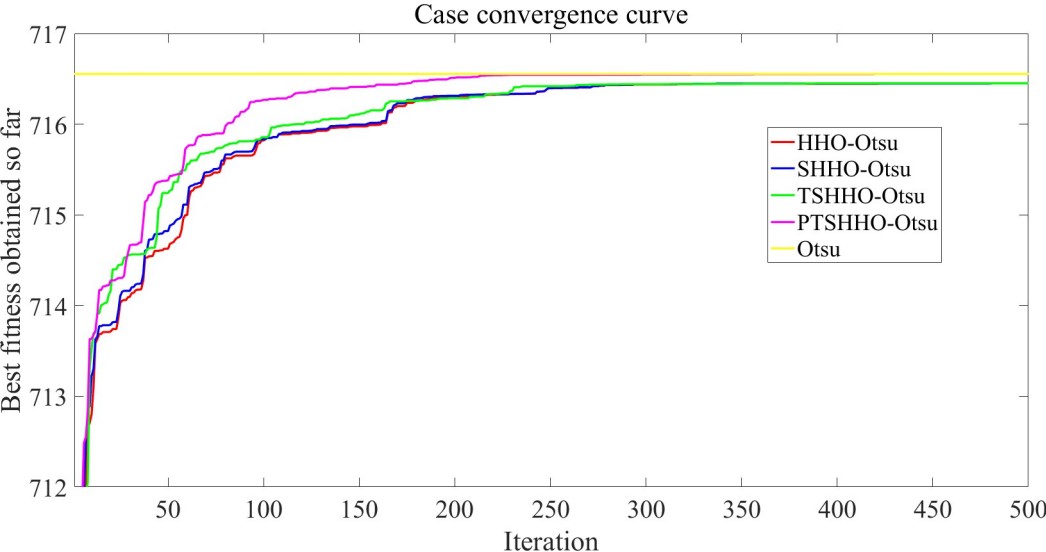

**Fig 8. Case convergence curve.**

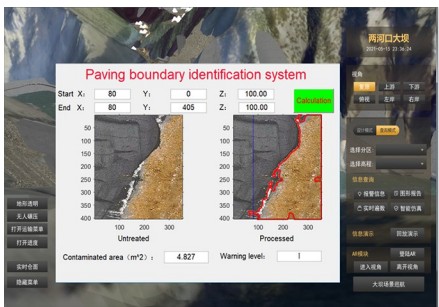 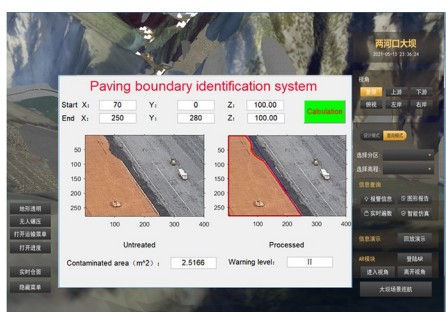 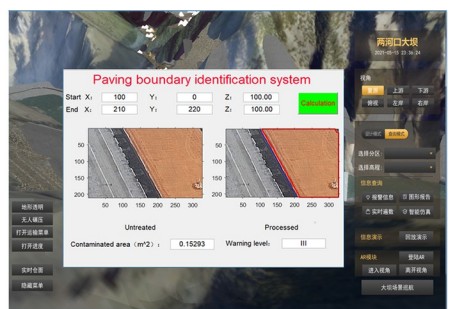

<div align="center">

**a:Level I  warning**          **b:Level II warning**          **c:Level III  warning**

</div>

**Fig 9.  Application system.**

is less than 4 m$^2$ but more than 1 m$^2$. A level III early warning is generated when the zone area is less than 1 m$^2$.

In addition, the operating speed of the system is approximately 20 s, which fully meets the construction requirements of the project site. The online operation of the system has been highly praised by construction sites because of its significant practical engineering value and suitability for other engineering fields.

## 5. Conclusion

This study optimizes the HHO algorithm, which significantly improves its accuracy and convergence speed. The universality, superiority, and consistency of PTSHHO are verified through multiple UM and MM test problems. In addition, PTSHHO and Otsu are combined to rapidly detect the boundaries of materials in high-definition images. Finally, the proposed method is applied to engineering data, and it achieves excellent results. The method is applied to a real-life project, where it reduces the calculation time to 20 s, which is approximately 18.8% of the original calculation time.

The proposed method can also be to other projects of the same type. Similar engineering application scenarios, such as roads and airports, must be verified in future engineering practices. In future work, other application scenarios for optimized PTSHHO can be developed and combined with other algorithms.

## Supporting information

**S1 Appendix.**
(DOCX)

## Author Contributions

**Conceptualization:** Xiaofeng Qu, Jiajun Wang, Xiaoling Wang.

**Data curation:** Xiaofeng Qu, Yike Hu.

**Formal analysis:** Xiaofeng Qu, Jiajun Wang, Xiaoling Wang.

**Funding acquisition:** Jiajun Wang, Xiaoling Wang.

**Investigation:** Xiaofeng Qu, Jiajun Wang, Yike Hu.

**Methodology:** Xiaofeng Qu, Jiajun Wang, Yike Hu, Dong Kang.

**Project administration:** Xiaofeng Qu, Jiajun Wang.

**Resources:** Xiaofeng Qu, Jiajun Wang, Xiaoling Wang.

**Software:** Xiaofeng Qu.

**Supervision:** Xiaofeng Qu, Jiajun Wang, Xiaoling Wang.

**Validation:** Xiaofeng Qu, Jiajun Wang.

**Visualization:** Xiaofeng Qu, Jiajun Wang.

**Writing – original draft:** Xiaofeng Qu, Jiajun Wang, Tianwen Tan.

**Writing – review & editing:** Xiaofeng Qu, Jiajun Wang.

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
