## [Decision Letter · Decision Letter 0]

30 Mar 2022

PONE-D-22-03955Fast Detection of Dam Zone Boundary Based on Otsu Thresholding Optimized by Enhanced Harris Hawks OptimizationPLOS ONE

Dear Dr. Wang,

Thank you for submitting your manuscript to PLOS ONE. After careful consideration, we feel that it has merit but does not fully meet PLOS ONE’s publication criteria as it currently stands. Therefore, we invite you to submit a revised version of the manuscript that addresses the points raised during the review process.

We look forward to receiving your revised manuscript.

Kind regards,

Seyedali Mirjalili

Academic Editor

PLOS ONE

Journal Requirements:

"This research was funded by the Yalong River Joint Funds of the National Natural Science Foundation of China (grant number U1965207) and the Young Scientists Fund of China (grant number 52009089)"

We note that you have provided funding information. However, funding information should not appear in the Acknowledgments section or other areas of your manuscript. We will only publish funding information present in the Funding Statement section of the online submission form. 

"This research was funded by the Yalong River Joint Funds of the National Natural Science Foundation of China (grant number U1965207) and the Young Scientists Fund of China (grant number 52009089).The funders had no role in study design, data collection and analysis, decision to publish, or preparation of the manuscript."

Reviewers' comments:

Reviewer's Responses to Questions

**Comments to the Author**

1. Is the manuscript technically sound, and do the data support the conclusions?

Reviewer #1: No

Reviewer #2: Yes

2. Has the statistical analysis been performed appropriately and rigorously? 

Reviewer #1: No

Reviewer #2: Yes

3. Have the authors made all data underlying the findings in their manuscript fully available?

Reviewer #1: No

Reviewer #2: Yes

4. Is the manuscript presented in an intelligible fashion and written in standard English?

Reviewer #1: No

Reviewer #2: Yes

5. Review Comments to the Author

Reviewer #1: In this paper, an enhanced version of the recently proposed HHO is proposed while the well-known PSO approach is also utilized for implementing some aspects into HHO. In my opinion, this paper is way beyond a low-level scientific paper and I recommend reject for this paper. However, the authors can use the following comments for updating their work in the future.

1- Why HHO and no other approaches. This algorithm is a new one so if you want to improve it, you have to compare it to the standard version of the HHSO and utilized the same problems that have been investigated before. Then you can use the standard and improved HHO for your specific field.

2- Too many references are cited. What are the differences of your work with others?

3- Is your work novel? Are there any other related work in the literature that worth mentioning in sec. 2? Please write about the novelty of your work. What are the main challenges in your field?

4- Using complex mathematical equations does not guarantee the acceptance of a paper. Reduce them and use them alongside some illustrations to make the paper readable by not specialists.

5- Quality of the illustrations are low. Not acceptable for a paper for journals

6- Figure 2. I can’t read and see what are the results and the differences. Only blue and red lines are visible

7- Figure 3. Convergence history should be visible

8- Why did you bring pseudo code after the results? The paper should be organized properly

9- Since the results of the proposed algorithm are not compared to the results of other algorithms, other well-known metaheuristics should be used for having a complete comparative investigation should be conducted by consideration of the results of other recently proposed metaheuristic algorithms:

10- The optimization problem statement in this paper is not in an acceptable level for a journal paper. A new section for this purpose is required.

11- The conclusion and the Abstract of the paper lacks the main quantitative results of this manuscript. It should be noted that some numerical results of the paper should be mentioned properly in the conclusion section.

12- Number of the Objective Function Evaluation (OFE) in the utilized methods and the modified version alongside the other methods in the revision should be provided in separate table for comparative purposes.

13- Improve the English level of the paper and correct the typos.

Reviewer #2: In this study a hybrid optimization algorithm based on the combination of HHO and PSO has been proposed to fast detection of dam zone boundary. The overall format of the paper is good and it is written in an organized manner. Although it is necessary to observe a few basic points and some major corrections, before I propose it for acceptance.

1- I suggest you combine the introduction and literature review parts. Also, the literature review part needs to be enriched with more references in … In recent years, many swarm optimization algorithms have been proposed and widely applied, such as particle swarm optimization [30], grey wolf optimizer [31]and Harris hawks optimizer [14]. Heidari et al. [14] compared HHO with other algorithms and demonstrated its advantages. HHO has proven its superiority in many applications [32-36], I suggest the following [1-3]:

• [1]. Reliability analysis based improved directional simulation using Harris Hawks optimization algorithm for engineering systems. Engineering Failure Analysis, 135, 106148. doi.org/10.1016/j.engfailanal.2022.106148

• [2]. First-order reliability method based on Harris Hawks Optimization for high-dimensional reliability analysis. Structural and Multidisciplinary Optimization, 62(4), 1951-1968. doi.org/10.1007/s00158-020-02587-3

• [3]. Using particle swarm optimization algorithm to optimally locating and controlling of pressure reducing valves for leakage minimization in water distribution systems. Sustainable Water Resources Management, 6(4), 1-11. doi.org/10.1007/s40899-020-00426-3

2- I would suggest showing the methodological approach through a flowchart.

3- Figs. 3 and 4 show the convergence curves of variant version of HHO. You can view the results of new algorithms such as AHA [4] or GNDO [5] (and etc.) and compare them with the results of your algorithm.

[1]. Artificial hummingbird algorithm: A new bio-inspired optimizer with its engineering applications. Computer Methods in Applied Mechanics and Engineering, 388, 114194. doi.org/10.1016/j.cma.2021.114194

[2]. Generalized normal distribution optimization and its applications in parameter extraction of photovoltaic models. Energy Conversion and Management, 224, 113301. doi.org/10.1016/j.enconman.2020.113301

4- In section 4, more details about case study is needed.

5- Conclusion has to be improved.

Good luck.

6. PLOS authors have the option to publish the peer review history of their article (what does this mean?). If published, this will include your full peer review and any attached files.

Reviewer #1: No

Reviewer #2: No

---

## [Author Response · Author response to Decision Letter 0]

19 May 2022

Thank you very much for your constructive comments. Your suggestion is of great help to my manucript. I have responded in detail to your comments in turn. According to your suggestions, we compare and analyze the results of PTSHHO, AHA and GNDO. It shows that PTSHHO is more suitable for complex and changeable sites and can converge well in most cases.For details, please refer to revised manuscript with track changes

---

## [Decision Letter · Decision Letter 1]

6 Jul 2022

Fast Detection of Dam Zone Boundary Based on Otsu Thresholding Optimized by Enhanced Harris Hawks Optimization

PONE-D-22-03955R1

Dear Dr. Wang,

We’re pleased to inform you that your manuscript has been judged scientifically suitable for publication and will be formally accepted for publication once it meets all outstanding technical requirements.

There is a comment on the quality of figure. Please make sure to improve the resolution in the proof. 

Kind regards,

Seyedali Mirjalili

Academic Editor

PLOS ONE

Additional Editor Comments (optional):

Reviewers' comments:

Reviewer's Responses to Questions

**Comments to the Author**

1. If the authors have adequately addressed your comments raised in a previous round of review and you feel that this manuscript is now acceptable for publication, you may indicate that here to bypass the “Comments to the Author” section, enter your conflict of interest statement in the “Confidential to Editor” section, and submit your "Accept" recommendation.

Reviewer #1: All comments have been addressed

Reviewer #2: All comments have been addressed

2. Is the manuscript technically sound, and do the data support the conclusions?

Reviewer #1: Yes

Reviewer #2: Yes

3. Has the statistical analysis been performed appropriately and rigorously? 

Reviewer #1: No

Reviewer #2: Yes

4. Have the authors made all data underlying the findings in their manuscript fully available?

Reviewer #1: No

Reviewer #2: Yes

5. Is the manuscript presented in an intelligible fashion and written in standard English?

Reviewer #1: Yes

Reviewer #2: Yes

6. Review Comments to the Author

Reviewer #1: The authors responded to my comments; however, the quality of figures are not in an acceptable level. They are still hard to recognize

Reviewer #2: (No Response)

7. PLOS authors have the option to publish the peer review history of their article (what does this mean?). If published, this will include your full peer review and any attached files.

Reviewer #1: No

Reviewer #2: No

---

## [Editor Report · Acceptance letter]

20 Jul 2022

PONE-D-22-03955R1 

Fast Detection of Dam Zone Boundary Based on Otsu Thresholding Optimized by Enhanced Harris Hawks Optimization 

Dear Dr. Wang:

I'm pleased to inform you that your manuscript has been deemed suitable for publication in PLOS ONE. Congratulations! Your manuscript is now with our production department. 

Kind regards, 

on behalf of

Prof. Seyedali Mirjalili 

Academic Editor

PLOS ONE